# Meeting Summary: Exploring Cloud Dynamics with Cloud Model 1 and 3D Visualization – insights from a University Modeling Workshop

Lisa Schielicke[1], Yidan Li[1], Jerome Schyns[1], Aaron Sperschneider[1], Jose Pablo Solano Marchini[1], and Christoph Peter Gatzen[2]

[1]University Bonn, Institute of Geosciences, Department Meteorology, Auf dem Hügel 20, 53121 Bonn, Germany
[2]European Severe Storms Laboratory e.V. (ESSL), c/o DLR, Münchener Str. 20 82234 Wessling, Germany

**Abstract.** We introduce an innovative two-week educational block course held at the University of Bonn during the 2023 winter semester, focusing on Cloud Model 1 (CM1) and its convection-resolving capabilities. During the course, participants gained essential skills in setting up and customizing CM1 simulations on a high-performance computing cluster, while gaining insights into moist convection dynamics. An additional introduction to three-dimensional visualization software allowed the participants to transform numerical data into compelling visualizations, deepening their insights into cloud dynamics. The participants applied their gained knowledge in research projects of their own choice that will be presented here and in the supplementary material.

## 1  Motivation

Thunderstorms hold a unique fascination for students of the atmospheric sciences. At the same time, numerical modeling plays a vital role in meteorology (e.g. Coiffier, 2011). Numerical weather prediction is a typical example of a high performance computing (HPC) application, commanding substantial computational resources (Vourlioti et al., 2023). Leading meteorological research and forecast centers such as the European Centre for Medium-Range Weather Forecasts (ECMWF), the National Center for Atmospheric Research (NCAR), the German Weather Service (Deutscher Wetterdienst, DWD), the Swiss National Supercomputing Centre (CSCS), MeteoSwiss use HPC systems to provide high-resolution weather (and climate) forecasts and reanalysis data (Nakaegawa, 2022). Hence, skills regarding the set-up, simulation, and analysis of numerical weather (and climate) models on HPC clusters are often essential for many career paths in meteorology. Here, we present an innovative solution that combines the development of these skills with the motivating topic of deep moist convection.

In this work, we will introduce an elective, 10-day intensive introductory modeling block course of Cloud Model 1 (CM1) accompanied by an introduction to a three-dimensional (3D) visualization software. It is intended for atmospheric sciences graduate students at the master's level and advanced undergraduate students from the fifth semester onward with an interest in cloud processes, thunderstorms, and 3D meteorological data visualization. Prior basic programming experience in e.g. Fortran, Python or R, familiarity with fluid dynamical equations and climate/weather models are expected at this level of education.

The course covers specific topics that are not covered by other courses at the University of Bonn (see Table 1 for an overview over the course topics). The learning objectives of the course are:

– Participants gain insight into the structure of a convection-resolving numerical weather model (Cloud Model 1).

   – They learn how an HPC works and actively run simulations on an HPC. They learn essential tools, e.g. schedulers (Slurm)

   – Participants learn to initiate, modify, visualize, and evaluate their own simulations. Modifications to the model are implemented using idealized experiments as examples, thereby, practicing using a high-level programming language.

– Participants will learn 3D visualization techniques to interpret model output, enabling them to effectively communicate their findings and insights.

   – Participants are able to adjust and run the model for a self-selected research question and present their results.

Through the active and intensive work with the model, participants gain insights into numerical weather modeling and will develop a foundational understanding, including the processes involved in modeling and the significance of model customization
in research. Many of the skills that participants acquired during the course can be applied broadly in meteorology beyond the specific model.

The block course introduces CM1, which is a state-of-the-art, convection-resolving, non-hydrostatic numerical model that allows the study of atmospheric phenomena in idealized setups (Bryan and Fritsch, 2002; Bryan, 2021). CM1 can be run on single as well as multi processors with grid spacings ranging from tens of kilometers down to a few meters (e.g. Orf et al.,
2017, who documentd a simulation of a tornado with 30 m grid spacing). It is available for download free of charge and its license allows the users to modify the source code according to their problems. It has been used to study i.a. tropical cyclones, mesoscale convective systems, supercells and thunderstorms, but also non-hydrostatic mountain waves, sea-breezes and low-cloud systems in an idealized way (see Bryan, 2021, for an extensive list of publications citing CM1). In comparison to operational models it is possible to study cloud systems in an idealized way, with systematic changes to the model environment
such as topography and atmospheric parameters. CM1 is commonly used in severe convective storms research, predominantly in the U.S., and a large scientific community is available for questions regarding the code, and exchange and discussion of results. CM1 is an ideal model for mastering the fundamental essentials required for model set-up, initialization, numerical integration, and output generation. Furthermore, it can serve as a valuable introduction to more complicated and realistic models such as the Weather Research and Forecasting Model (WRF), the Consortium for Small-scale Modeling (COSMO)
model, the Icosahedral Nonhydrostatic Weather and Climate (ICON) Model and others.

Next to the introduction to CM1, the block course set a focus on visualizing the results. Data visualization is indispensable in atmospheric science. With help of weather charts at different pressure levels, cross sections and vertical profiles, meteorologists explore the atmosphere's three-dimensional state for informed forecasts. Additionally to two-dimensional (2D) weather charts, three-dimensional (3D) visualization aids in understanding intricate processes like flow dynamics and development.

Rautenhaus et al. (2017) gives an overview over visualization techniques commonly used in operational weather forecasting and in meteorological research and emphasize its importance for – among others – visual mapping of observations and simulation, flow analysis and its temporal evolution, the detection and tracking of atmospheric features as well as the analysis of uncertainty in simulations. Although very important, the topic of 3D visualization is rarely covered in lectures of meteorology. Since 3d visualization has the potential to give additional insights into the dynamics of atmospheric phenomena, we covered this topic in the course, too. We focused on the open source 3D visualization software ParaView for the 3D visualization since it is available at the HPC cluster Bonna of University Bonn. Additionally, ParaView is well documented with extensive online learning material (ParaView Developers, 2020). Bonna is a Massively Parallel Processor (MPP) cluster with 70 nodes, each with dual Intel Xeon Gold 6130 CPU processors which operates on CentOS Linux with a Slurm batch system.

In this work, we want to advertise the course material and present the outcome of the modeling course to a greater community. The idea is that this paper can help to create own model workshops based on the freely-available course material. Moreover, we want to share results of the participant's projects that showcase the applicability of the course material to realize own research ideas. This paper is organized as follows: A course overview is given in section 2. The success of the course is evaluated by a survey that assess the knowledge before and after the course. Some results of the surveys are presentend in section 3. A brief overview over the participant's projects is presented in section 4 and in greater detail in the supplementary material. A summarizing conclusion is given in section 5.

## 2 Course design and course requirements

The course, entitled "2-weeks block course: Cloud Model 1 (CM1)", was scheduled for a 10-day period (2 weeks, Monday to Friday, 10 am to 4 pm) and was given partially in person, partially online. The course covered vital topics from model compilation, HPC operation, code modification to data visualization within the two weeks. In the first week, the participants learned the essentials of running and visualizing CM1 simulations on an HPC, applied to specific problems such as the simulation of a supercell. In the subsequent week the participants mainly worked on independent small-scale research projects, building upon their acquired knowledge. The course intended to show various possibilities to use CM1. It started by simple application of pre-configured cases and ended by (simple) changes implemented in the code. Topics always started from "easy" to "advanced": from setting-up and running CM1 on an HPC cluster to the simulation pre-configured test cases of CM1 via modified namelists to the implementation of changes to the source code and the use of external terrain. Participants learned not only the basics of how to work with a numerical model but also how to get started and continuously improve their modeling skills. Refer to Table 1 for comprehensive course details and the timetable. The course material is published under a Creative Commons License as open educational resource (OER Schielicke, 2024). The course content was mainly delivered as active learning activities. The amount of frontal lectures given by the instructor was at a minimum. Participants worked individually and in their own time on the course material. Each of the participants had access to a Google document of the course material which was successively extended day-by-day. They were asked to take notes, add comments and answer questions directly in their

**Table 1.** Overview over the course topics and timetable of the 2 weeks CM1 and visualization modeling course.

| Time | Day 01 | Day 02 | Day 03 | Day 04 | Day 05 |
|---|---|---|---|---|---|
| Morning | HPC, Slurm, Vim, scp/ssh | Compiling CM1 and first (pre-configured) supercell simulation | Introduction to ParaView (continued); Visualization of supercell data | External sounding, supercell literature, setting umove and vmove in namelist (moving domain) | Changes to the source code (continued); Presentation and discussion of results (so far) |
| Afternoon | CM1 source code, CM1 website | Introduction to Para-View | CM1 governing equations; Namelist | Changing the appearance of the warm bubble in the CM1 source code | Terrain (idealized) |

| Time | Day 06 | Day 07 | Day 08 | Day 09 | Day 10 |
|---|---|---|---|---|---|
| Morning | Terrain (continued); Real terrain | Restarts; How to deal with error messages | Project | Project | Project |
| Afternoons | Real terrain (continued) | Presentation of own project idea and rough project plan | Project | Project | Updated presentation of own project (state up to this day); Planning of the final report |

Google document. The instructor had access to these documents and could monitor the progress of each participant. In case of problems, the instructor gave guidance on how to solve them. Regular mentoring sessions of the whole group under guidance of the instructor provided opportunities for feedback and progress discussion. At some times, the students helped each other to complete tasks, for example for visualization solutions. They also helped each other in finding a balance regarding the extent of simulations done for the research topic.

For the successful completion of the course, each participant had to attend and participate actively for at least 80% of the course time and had to hand in a report of about 20 pages four weeks after the course ended. The participants chose their own research topics for the report. A guideline for writing the report was given in the course material (Chapter Day 08-10). The reports had the main goal that the participants documented how to run experiments with CM1 on an HPC cluster and how to visualize the data. An extensive research on the literature background of the topics was not required, however, the participants were asked to – at least – read the cited literature that is provided in the specific test case folders or in the source code. The

reports were evaluated according to the guideline given in the course material. The final grade was given on the basis of the reports and the active participation during the course.

The course is a stand-alone course that was originally planned for master students, but also advanced bachelor students (5th or 6th semester) were encouraged to take the course as an extracurricular activity. In March 2023, four participants finished the course, two of them were still Bachelor students. Only one of the participants had prior experience with CM1. Generally, the course is best accompanied by a course in convective or mesoscale meteorology, as this aids in the interpretation of the produced data.

In this work, we use Cloud Model 1, version cm1r21.0 (released 20 April 2022), which was the most current version available during the workshop. We compiled CM1 to enable obtaining output in a netCDF format, which facilitates subsequent analysis. Note that the CM1 model simulations are generally initialized with a single vertical profile that determines the initial atmospheric conditions of the whole domain. Although the course was designed to set-up the model on a specific HPC cluster (Bonna of the University of Bonn), users of single-processor are encouraged to run through the course material, too. However, they are referred to the CM1 website to learn what is required and how to compile the model for single-processors. Unix/Linux environment makes it easier to use, although it is also possible to run CM1 on a Windows (or Mac) computer, too.

As visualization tools the focus lay on ParaView and Ncview, which were both already available at the HPC cluster Bonna of University Bonn. To get a fast overview over the data, the two-dimensional (2D) visualization tool Ncview proved to be helpful as a start before further more computationally costing three-dimensional (3D) tools were used. Ncview is a visual browser that allows a fast examination of netCDF data (Pierce, 2021). On the other hand, ParaView is an open source 3D visualization software. It can be used to interactively analyse large data sets, but could also be used via scripts (Ahrens et al., 2005; Ayachit, 2015). It is well documented online and the participants had two half days to work through the online material (ParaView Developers, 2020). One of the participants had prior expertise in VAPOR (Visualization and Analysis Platform) of the National Center for Atmospheric Research's Computational and Information Systems Lab. VAPOR is another 3D visualization tool that can be used either interactively or via scripts to produce animations or images (Li et al., 2019; Visualization & Analysis Systems Technologies, 2023). It has similar advantages as ParaView with a broad user community and extensive online documentation.

Due to time constraints, we could not cover all topics. An incomplete list of topics not or only partially-covered are: (i) the Courant-Friedrichs-Lewy (CFL)-criterion, (ii) dealing with errors and error messages, (iii) testing more pre-configured test cases, (iv) releasing parcels, (v) restarts with higher resolution, (vi) exploring the difference between large-eddy simulations (LES) and direct numerical simulations (DNS), and so forth. A follow-up course is currently being planned.

## 3 Participants' feedback

The workshop had several learning goals for the participants. To see whether these learning goals were both meaningful to and achieved by the participants, pre- and post-assessments were conducted with the 4 participants that successfully completed the course. The overview given here is limited to programming skills. It is noticeable from the pre-assessment that the majority of the participants had little to no experience at the beginning of the course with the working environment and tools taught

| Topic | none | little | moderate | good | very good | No answer |
|---|---|---|---|---|---|---|
| CM1 | 1 (blue) | 2 (blue) | 1 (blue) | 4 (orange) | | |
| HPC | 4 (blue) | | | 2 (orange) | 2 (orange) | |
| Slurm | 4 (blue) | | | 3 (orange) | 1 (orange) | |
| CDOs | 2 (blue) | 1 (orange), 1 (blue) | | 2 (orange) | 1 (orange) | |
| Netcdf | | 2 (blue) | 1 (blue) | 1 (orange), 1 (blue) | 2 (orange) | 1 (orange) |
| Visualization | 1 (blue) | 1 (blue) | 2 (blue) | 2 (orange) | 2 (orange) | |
| ParaView | 3 (blue) | 1 (blue) | | 3 (orange) | 1 (orange) | |
| Python | | 1 (blue) | 2 (blue) | 2 (blue), 1 (orange) | 2 (orange) | |
| R | | 2 (blue) | 1 (orange), 1 (blue) | 2 (orange), 1 (blue) | | 1 (orange) |
| Fortran | 1 (blue) | 1 (blue) | 1 (orange), 2 (blue) | 1 (orange) | 1 (orange) | 1 (orange) |

**Figure 1.** Results of the survey question regarding the assessment of the prior skills of certain course topics before course start (blue dots on the left side of the columns) and their self-assessed skills after the course (orange dots on then right of the columns).

(Fig. 1). In particular, the participants had not dealt with HPC clusters before, which is one of the key competencies that is taught in the course. The situation was not much different in the area of (3D) visualization of simulation results, with none of the participants rating their own ability as "good". However, most of the participants were already familiar with programming languages such as Python and R when they started the course. After all, there was mostly little experience with the CM1 model used.

Following the course, a further survey was conducted. Students were asked to recap the course outcome 11 months after the course to give some more insights about the longer term benefit of the course, e.g. consider which skills they developed during the course that they are using since. Compared to the answers at the beginning of the course, there have been several changes. All participants now state that they have a good knowledge of CM1. The goal of becoming familiar with this model has clearly been fulfilled. The experience in working with HPC clusters, with Slurm and CDOs was predominantly rated as "moderate" and even "good". There is therefore reason to believe that the participants have gained more than just a first insight, so that further improvement of these skills is possible individually, though, it can not be ruled out that some of the post-processed results are biased with respect to additional knowledge gained within the 11 months after the course ended. It is interesting that the subjective assessment of the skills in programming and handling netcdf data has improved substantially, even though a certain level of experience was already available at the beginning of the course. This shows that this intensive workshop not only provided new specific skills, but was also a useful addition to the usual curriculum at the university. The surveys on the feedback of the participants could therefore clearly show that the goals set for the workshop were meaningful, and that these goals were also achieved. The complete survey is given in the supplementary material S1.

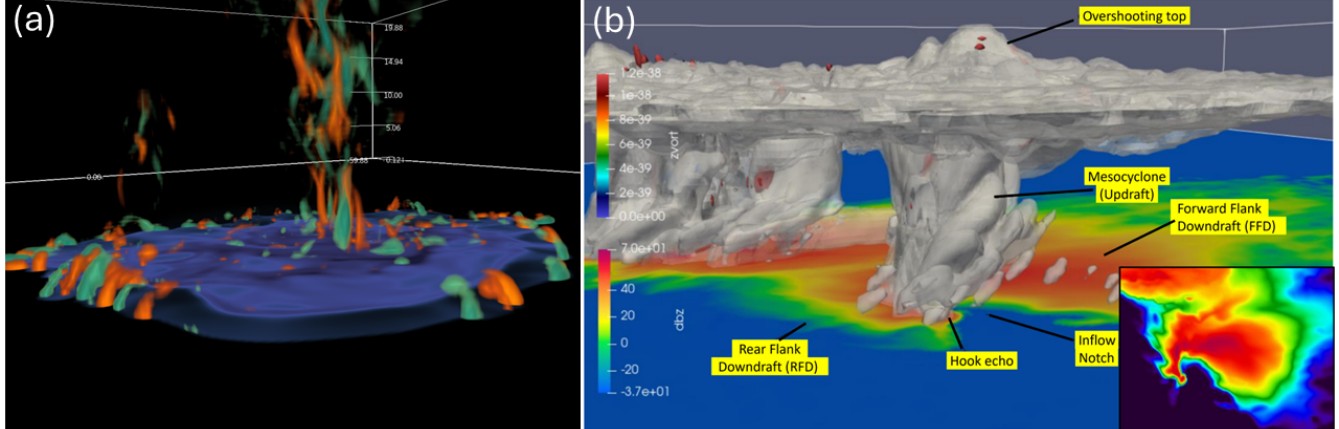

**Figure 2.** 3D-visualizations of projects of the participants: (a) Visualization of the cold pool formed by short lived cells in a weakly dynamic environment. Displayed are the negative potential temperature perturbation in the lowest model-level (blue, $-0.5$ K isotherm); and vorticity volumes (orange: $0.013s^{-1}$; turquoise: $-0.013s^{-1}$). Figure was plotted by Aaron Sperschneider using VAPOR. (b) 3D visualization of a right-moving supercell with its typical characteristics. Shown is the reflectivity at 250 m (in dBZ, color-shaded at the surface for the 250 m level; and in the inset (2D plot) at 750 m). Whitish colors represent the cloud composed of the mixing ratios of cloud particles ($q_c$), graupel ($q_g$) and ice ($q_i$). Inside the clouds, colored shadings represent positive vertical vorticity $\zeta$ surfaces with values of $\zeta = 0.03s^{-1}$ and $\zeta = 0.05s^{-1}$. Figure was plotted using ParaView (3D) and ncview (2D) by Jerome Schyns.

## 4 Participants projects

In this section we will introduce some research projects that were part of fulfilling the course requirements. A detailed description of the projects including two- and three-dimensional visualizations can be found in the supplementary material S2.

- **Influence of topography on deep moist convection under weak lifting conditions:** The participant studied how local topography impacts intense storms under weak lifting conditions, particularly over lower mountain ranges. Visualization software highlighted storm structures and vorticity patterns, offering practical insights for nowcasting and emphasizing local topography's role. Figure 2(a) illustrates downdrafts' connection to vorticity patches, enriching the study. See suppl. material, section S2.1, for more details.

- **Study of different lifting processes and modified soundings of the Moore/Oklahoma tornado outbreak on 3 May 1999:** Using CM1, the study investigated various initiation setups and lifting mechanisms, underlining the need to modify soundings derived from reanalysis for accurate outcomes. Additionally, high-resolution simulations were conducted to visualize the three-dimensional structure and characteristics of a supercell (Fig. 2(b)). The project shows the importance of initiation mechanisms and environmental parameters in influencing convective modes and appearances. See suppl. material, section S2.2, for more details.

– **Comparing the development and microphysical composition of high-humidity/classic/low-humidity supercells:** The project investigates supercell behavior in varied moisture environments with and without terrain consideration, using CM1 simulations and modified Weisman-Klemp supercell soundings (Weisman and Klemp, 1982). Findings emphasize the influence of initial humidity on supercell microphysical composition and development, as well as the role of the terrain. Overall, the study highlights how environmental factors affect storm development and microphysical processes, offering valuable insights for research and education. See suppl. material, section S2.3, for more details.

– **Visualization of precipitating and non-precipitating shallow cumulus:** Using ParaView, different cloud and rain mixing ratio thresholds were explored for the visualization of shallow cumulus clouds and precipitating processes. This project highlights the importance of careful visualization and technical considerations for effective data representation. The simulations demonstrated that cloud cover thinned during stronger precipitation, aligning with previous findings (VanZanten et al., 2011). Overall, the project provided educational insights into shallow cumulus clouds and their evolving composition. See suppl. material, section S2.4, for more details.

## 5 Conclusions

This publication presents results from a 2-week modeling workshop at the University of Bonn in March 2023, focusing on running and modifying Cloud Model 1 (CM1) and utilizing 3D visualization software (see Schielicke, 2024, for course material). Participants learned essential skills through hands-on examples and worked on self-selected research questions during the second week, documenting their skills in course reports. The active learning approach in the course encouraged participants to engage directly with tasks related to topics such as HPC, Slurm, CM1 and data visualization. Participants worked independently on assigned tasks, receiving guidance from the instructor when needed. Regular discussion and mentoring sessions provided opportunities for feedback and progress discussion. Pre- and post-course surveys indicated substantial improvement in participants' skills and confidence levels, demonstrating the effectiveness of the active learning methodology(see section 3 and supplementary material section S1). High-resolution idealized modeling highlighted key aspects of convective processes. Data visualization helped to deepen the understanding of convective processes. Participants became increasingly independent over time, exploring various research topics and visualization techniques. The course's real-world setting and use of open-source software offer valuable experience applicable to future careers. Future improvements include extending error management time and integrating lectures on mesoscale dynamics.

Over time, the participants became increasingly independent. They engaged deeper in their specific research topics and explored multiple ways to visualize their results. In addition, the course teaches useful skills that can be applied in their thesis preparation, graduation phase or in a later professional research career. This additional benefit has also been documented in other disciplines that use active learning strategies in teaching (Wilke, 2003) and is successfully used in atmospheric sciences, too (Handlos et al., 2022; Steeneveld and de Arellano, 2019). In contrast to active learning methods that rely on software designed exclusively for educational purposes (Limbach et al., 2015), the strength of this course lies in its ability to offer students a scientific setting, mirroring possible challenges they might face in their future careers. Furthermore, openly accessible

(open-source) software is used, encouraging participants autonomy in conducting independent research without depending on software solutions that are available at specific institutes only. An analogy to this approach is to provide them with Lego building blocks, enabling the construction of a model car tailored to their particular (research) requirements, rather than providing a pre-assembled vehicle.

To summarize, the presented CM1 course teaches important skills regarding model setup and simulation and serves as a basic introduction to model code in general, which lays the groundwork for tackling more realistic models, for example WRF, COSMO or ICON. The introduction to 2D and 3D visualization software such as Ncview, ParaView, and VAPOR, complements the learning outcome. The work on projects of the participant's own choice further motivates and reinforces the application of learned content. Finally, the participants actively use the CM1: two students used the model in their bachelor thesis, another participant used CM1 and 3D visualization techniques in a research contribution at a conference (Sperschneider and Bott, 2023).

*Code availability.* The course contents are available as open educational resource (OER, see Schielicke, 2024). The CM1 source code is available at the CM1 website (Bryan, 2021).

*Author contributions.* LS planned and structured the paper and designed the course material; AS, JS, JPSM and YL designed their own research projects, generated and visualized the data associated with their subsection, JPSM and YL provided a concluding statement (in the supplemental material, section S1.3), CG and LS wrote and revised the text of the manuscript and the summaries of the projects based on AS's, JS's, JPSM's, YL's course reports in equal shares. All authors discussed the manuscript continuously to improve the text.

*Competing interests.* We declare that no competing interests are present.

*Acknowledgements.* The authors gratefully acknowledge the granted access to the Bonna cluster hosted by the University of Bonn along with the support provided by its High Performance Computing & Analytics Lab. LS would like to thank Ingo Kirchner (FU Berlin, Germany) for providing a script that facilitates the compilation of CM1 on an HPC cluster. We extend our gratitude to OpenAI's GPT-3.5-based language model for its assistance in improving the use of the language as well as refining parts of the text. Furthermore, it helped in drafting summaries of the student reports. Finally, we would like to thank two anonymous reviewers and the editor Heini Wernli whose comments helped us to improve the manuscript.

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
