# Peer review of "Meeting Summary: Exploring Cloud Dynamics with Cloud Model 1 and 3D Visualization – insights from a University Modeling Workshop"

_EGUsphere, 2023_

## Referee Comment (RC2)

**Recommendation**: Minor revisions

**Summary**: The paper outlines a 2-week block course designed to teach participants how to run the CM1 model and use state-of-the-art visualization tools to analyze the data.  Detailed lesson plans are also provided, enabling instructors to easily adopt the material.  The paper is well-written, and I only have a handful of minor suggestions.

**General comments:**

1. Can the authors provide a URL pointing directly to the course material?  I landed on a German webpage that I found hard to navigate (I struggled a bit finding the course content).

2. Perhaps use "participant" rather than "student"?

**Specific comments:**

1. Line 2: CM1 is actually convection *resolving* (not only convection *permitting*).

2. Line 10: Suggest adding "e.g." before the reference.

3. Line 21: Use state-of-the-art instead of up-to-date? And again, consider "cloud resolving" instead of convection permitting.

4. Line 25: run → documented?

5. Lines 48, 49: Replace "chapter" with "section"

6. Line 55: Did the participants have to produce the 20 page report within the 2-week period?

7. Line 67: Here and elsewhere: Instead of single-computer, consider "single-processor" or "serial application"

8. Line 80:  Is this level of detail needed (background of individual participants)?

9. Line 81: Suggest rewording: Not or only partially covered.

10. Line 94: How did the participant quantify the degree of lift in the field?

11. Line 104: Higher-resolution: Compared to what—perhaps report the grid spacing used in the different simulations?

12. Line 111, section header: Instead of "classic" perhaps use "intermediate"?  The term "classic supercell" is usually part of the "high-precipitation", "classic", "low-precipitation" trio.

13. Line 125: Add a reference here?

14. Line 157: Replace "Fazit" with "feedback", and consider removing "and co-author to this paper"

---

## Author Comment (AC2)

**Reply to Reviewer #2**

We thank Reviewer #2 for the helpful comments! Please note that we will change parts of the paper based on the suggestions of Reviewer #1: We would like to reorganize parts of the paper. We will add more details regarding the course background, design, requirements and goals in the introduction. Additionally, we used a survey to ask the participants of the first course to recap what they have learned. These results will be added to the conclusion part. To stay within the recommended page range, we will shorten the part about the students' contributions and write it in a more concise way. As a result of these changes, some of your comments might not be fully addressed in the final paper. However, we greatly appreciate your valuable feedback, which has contributed to improving the quality of our paper.

Please note that we formatted the reviewer comments in bold and cursive text settings, and the authors' answers in blue.

***Comments of Reviewer #2 (abbreviated as R2): Recommendation: Minor revisions***

***RC2: Summary: The paper outlines a 2-week block course designed to teach participants how to run the CM1 model and use state-of-the-art visualization tools to analyze the data. Detailed lesson plans are also provided, enabling instructors to easily adopt the material. The paper is well-written, and I only have a handful of minor suggestions.***

> Answers and Comments of the Authors (abbreviated as AC): Thanks for your helpful comments! We will address you comments point by point in the following.

**General comments:**

***RC2: 1. Can the authors provide a URL pointing directly to the course material? I landed on a German webpage that I found hard to navigate (I struggled a bit finding the course content).***
> AC: Thanks for the advice! We additionally uploaded the material to researchgate (under the same license), so it should be easier accessible for non-german speakers. The reference is: Schielicke, Lisa, 2024: "Cloud Model 1 & Visualization - A Block course", ResearchGate, Preprint, January 2024, last access: 2-Feb.2024. DOI: 10.13140/RG.2.2.30017.12642

***RC2: 2. Perhaps use "participant" rather than "student"?***
> AC: Thanks, we will change it according to your suggestion.

***RC2: Specific comments:***

***RC2: 1. Line 2: CM1 is actually convection resolving (not only convection permitting).***
> AC: Thanks, we will use convection-resolving instead

*RC2: 2. Line 10: Suggest adding "e.g." before the reference.*
    AC: Thanks, we will do this.

*RC2: 3. Line 21: Use state-of-the-art instead of up-to-date? And again, consider "cloud resolving" instead of convection permitting.*
    AC: We will follow your advice.

*RC2: 4. Line 25: run  documented?*
    AC: Thanks, we will change "run" to "documented" in line 24.

*RC2: 5. Lines 48, 49: Replace "chapter" with "section"*
    AC: Thanks, we will replace chapter with section.

*RC2: 6. Line 55: Did the participants have to produce the 20 page report within the 2-week period?*
    AC: No, they had approximately 4 weeks to finish the report. We will clarify this in the text.

*RC2: 7. Line 67: Here and elsewhere: Instead of single-computer, consider "single-processor" or "serial application"*
    AC: Thanks! We will change it according to your suggestion to single-processor.

*RC2: 8. Line 80: Is this level of detail needed (background of individual participants)?*
    AC: Thanks, we will delete the extra information.

*RC2: 9. Line 81: Suggest rewording: Not or only partially covered.*
    AC: Thanks, we will follow your advice.

*RC2: 10. Line 94: How did the participant quantify the degree of lift in the field?*
    AC: Thank you for the comment - indeed our text can be more clear here: With "weaker" forcing, it is meant that storm cell intensity is regarded in a thermodynamic profile associated with a weak forcing scenario. This scenario was characterized by the absence of quasi-geostrophic forcing as well as the absence of frontal lift. As a measure to investigate cell intensity, the participant used the maximum vertical velocity of each cell. In a later work based on the presented work during this course, the participant compared his results to a strongly-forced situation using a sounding associated with frontal lift on another day:

    REFERENCE: Sperschneider, A. and Bott, A.: Influence of the Orography of West-Central European Low Mountain Ranges on the Intensity of Deep Moist Convection, 11th European Conference on Severe Storms, Bucharest, Romania, 8–12 May 2023, ECSS2023-89, https://doi.org/10.5194/ecss2023-89, 2023.

*RC2: 11. Line 104: Higher-resolution: Compared to what—perhaps report the grid spacing used in the different simulations?*
    AC: The majority of the simulations had horizontal grid spacings of 1000 m and a vertical grid spacing of 500 m. In the high-resolution run the horizontal grid spacings

changed to 500 m. We will clarify this in the text and add the grid spacing to the figure..

**RC2: 12. Line 111, section header: Instead of "classic" perhaps use "intermediate"? The term "classic supercell" is usually part of the "high-precipitation", "classic", "low-precipitation" trio.**

AC: Probably the text is not clear enough. Indeed, we focus on a classic supercell structure (Weisman-Klemp sounding) and investigate if it evolves into a high-precip or low-precip storm in response to modifications of the environmental moisture profile. We will clarify this point in the manuscript accordingly. However, it might change due to the changes we plan to make in this chapter.

**RC2: 13. Line 125: Add a reference here?**

AC: Thanks, we will add a reference here.

**RC2: 14. Line 157: Replace "Fazit" with "feedback", and consider removing "and co-author to this paper**

AC: Thanks, we will change both according to your suggestions.

---

## Author Response (AR1)

**Changes done to the paper entitled Meeting Summary: Exploring Cloud Dynamics with Cloud Model 1 and 3D Visualization – insights from a University Modeling Workshop**

Author(s): Lisa Schielicke et al.

MS No.: egusphere-2023-2700

MS type: Meeting summary

In the following, we provide a detailed point-by-point response to all referee comments and specify all changes in the revised manuscript. The responses to the referees are structured as follows: (1) comments from referees are given in bold, cursive, black text and start with RC (followed by the number of the reviewer), (2) author's response in blue, normal text which starts with AR, (3) author's changes in manuscript given in pink, cursive text introduced by AC. In addition, we also provide a marked-up manuscript version showing the changes made generated by latexdiff.

**Changes done according to Reviewer #1 comments**

AR: We thank Reviewer #1 for the helpful comments. Based on the suggestions in the general comments part of Reviewer #1, we would like to reorganize parts of the paper. We will add more details regarding the course background, design, requirements and goals in the introduction. Additionally, we used a survey to ask the participants of the first course to recap what they have learned. These results will be added to the conclusion part. To stay within the recommended page range, we will shorten the part about the students' contributions and write it in a more concise way.

Please note that we formatted the reviewer comments in bold and cursive text settings, and the authors' answers in blue.

**Comments of Reviewer #1 (abbreviated as RC1):** *General comments:*

*RC1: 1. The motivation/introduction would be improved with additional information. For example, example, where in the curriculum does this course fit in? Are these undergraduate students or graduate students? Is there any other course that covers this material? Additional background about this and why the course was developed would be useful to include. Are there pre-existing courses elsewhere that this course was modeled after? The degree of innovation of this course would be clearer with both more specific context of the institution as well as broader community context (i.e., how novel is this course).*

Author Responses (abbreviated as AR) to the Reviewer Comments (RC1): The course was intended for graduate students at the master level and for advanced undergraduate students of meteorology or atmospheric sciences of semester 5 and upward. Basic programming experience with at least one programming language (such as Fortran, Python, or similar) is expected at this level of education. Additionally, the basic knowledge of the fluid dynamical equations and of what climate and weather models do is expected. The course is mandatory for students with an interest in cloud processes, thunderstorms and visualization of meteorological data. No other course at the University of Bonn covers the specific topics offered in the course. To our knowledge there are courses in Germany that teach the use of climate and numerical weather models, however we have seen no course specifically on the topic of (severe) convection. At first, numerical weather prediction and climate models may seem mysterious to students, akin to black boxes generating data. Through this block course, our goal is to demystify their operation, using CM1 as an example. Students will navigate from initialization to visualization, empowering them to customize code and methods for their own research. This leads to a deeper understanding of numerical weather modeling, while also equipping students to confidently conduct simulations on high-performance computing systems. The advantage of a course based on CM1 is that students can experiment freely with the code and idealized set-ups. This allows a deeper understanding of how a model behaves dependent on the chosen set-up like boundary layer conditions, resolution, and parameterization schemes. Of course, the model sometimes does not work as expected and it might be challenging to find the reasons and workarounds. Hence, this course offers a safe environment to explore failures as well and find solutions.

The course was developed partially because of personal interests of the main instructor. The author has a strong interest in (severe) convection herself and was fascinated by 3d animations/visualizations of supercells such as those done by Leigh Orf (https://orf.media/ , last access: 3-Feb-2024). One motivation for the students is that they will be capable of producing high-quality 3d images from thunderstorms. On the other hand, many students express a strong interest in thunderstorms and cloud processes and give this as a reason for studying meteorology. To meet these interests the course was developed. To our knowledge there is no specific course on convection-resolving models offered at German Universities. However, the topics learned are also general such as the usage of an HPC and slurm, using namelists to start simulations, or 3d visualizations. We think that this course offers learning topics beyond the specific model that can be useful in other areas of meteorology and beyond, too.

Additionally, the project topics chosen in the field of convective storms cover this field in detail that is beyond the general learning curriculum. It includes the forecast of convection initiation as a new framework that supports the conceptual understanding of how the atmosphere works and how numerical models deal with the representation of the associated processes.

*AC: We added some more text to the introduction to motivate the course.Some of the answers will be given in more detail in section 2 (see below). Note, the course is not mandatory; it is an elective course that supplements other courses given at the University of Bonn for students with an interest in convection, e.g. the Stormchasing course.*

*RC1: 2. With regards to course design/requirements, I have several questions. What was the title of the course? What were the course learning objectives? Did students self-select into this course? How many hours per day and days per week did the course meet? How did the authors decide the topics to cover? How were students evaluated? What was submitted with the project? What were the course requirements? Was the class conducted using in-person instruction? What was the method of content delivery (e.g., lecture vs. active learning activities vs. group work, etc)? How many students were enrolled? This section in particular needed a lot more information in it to give readers a good sense of what the course was like.*

AR: Thanks for the questions. I will answer them in the following:

**Course Title and Schedule**: The course is entitled "2-weeks block course: Cloud Model 1 (CM1)". We met for 10 days (2 weeks, Monday to Friday) between 10 am and 4 pm partially in person, partially online (partially due to train strikes in Germany).

**Learning objectives:** The overall learning objectives are the following:

- **Understand model initialization:** Students will comprehend the process of initializing complex models, including the selection of namelist variables, to gain insight into how models are set up and prepared for simulations.
- **Demystify model operation:** Students will learn to navigate and manipulate model code, transitioning from perceiving models as black boxes to understanding them as accessible tools for conducting simulations.
- **Model modification:** Students will gain the ability to modify model code according to their research questions and preferences, fostering adaptability and creativity in utilizing modeling tools.
- **Visualization:** Students will learn 3D visualization techniques to interpret model output, enabling them to effectively communicate their findings and insights.
- **Confidence in model usage:** Students will build confidence in utilizing models independently, beyond the confines of the course, by understanding their inner workings and operation on high-performance computing systems.
- **Improve understanding of numerical weather modeling:** Students will develop a foundational understanding of numerical weather modeling, including the processes involved and the significance of model customization in research.

The learning intended to show various possibilities to use CM1. It started by simple application of pre-configured cases and ended by (simple) changes implemented in the code. Topics always started from "easy" to "advanced": (1) Set-up and run of CM1 on a high-performance computing (HPC) cluster; (2) Simulate pre-configured test cases of CM1; (3) Adapt simulations by using modified namelists; (4) Implement changes to the source code; (5) Implement terrain. Participants learned not only the basics of how to work with a numerical model, but also how to get started and consecutively evolve the abilities.

**Course completion requirements:** For the successful completion of the course, students had to attend and participate actively for at least 80% of the course time and had to hand in a report of about 20 pages 3 weeks after the course. A guideline for writing the report was given in the course material (course material, Chapter Day 08-10, Schielicke, 2023, http://dx.doi.org/10.13140/RG.2.2.30017.12642). The research topics were chosen by the students themselves. Some had prior knowledge of severe weather

topics and already had ideas on what to study. They presented their ideas in the course and were guided through the development process. Others were directed to the extensive selection of test cases available in CM1 and were encouraged to reproduce one or more of the cases and document the process. The reports had the main goal that students documented how to run experiments with CM1 on a high-performance computing (HPC) cluster and how to visualize the data. An extensive research on the literature background of the topics was not required, however, the students were asked to – at least – read the cited literature that is provided in the specific test case folders or in the source code. The reports were evaluated according to the guideline given in the course material. The final grade was given on the basis of the reports and the active participation during the course (see below).

**Number of participants:** Five students were enrolled at first, but one of them became ill and couldn't complete the course.

**Instructional format and content delivery:** The content was mainly delivered as active learning activities, the amount of frontal lectures given by the instructor was at a minimum. Students worked individually and in their own time on the course material (see reference above). Each student had the course material as a google document and were encouraged to write down tasks and comments directly in their document. The instructor had access to the google documents, and hence could judge the active participation. In case of problems, the instructor gave guidance on how to solve them. At some times, the students helped each other to complete tasks, for example for visualization solutions. They also helped each other in finding a balance regarding the extent of simulations done for the research topic.

*AC: We added the learning objectives and more details regarding the course, instructional format, etc. to the introduction and added more text to section 2 "Course design and course requirements" of the manuscript.*

*RC1: 3. Highlighting the student projects is interesting, though given the limited time spent on them (thus preventing a more detailed analysis), I would be more curious about student feedback about the course and the skills that they were supposed to acquire. If the authors continue to include these projects, a bit more information about them would be helpful (e.g., how these were assessed). Alternatively, simply including them in the supplemental material would be sufficient.*

AR: We appreciate this comment that improves the manuscript! We will ask the students for additional feedback to include this in the text. It is true that the research projects fit in the supplementary material.

*AC: We did a survey with the course participants to assess their skills and knowledge, interests and expectations as well as feedback after the course. We present the results in the new section 3 "Participants' feedback"*

*RC1: 4. Was there any kind of pre- and post-assessment given to students? Being able to demonstrate that this course was impactful on student understanding of convection (in a quantified way) as well as specific skills would be very useful, particularly given the short timeline.*

AR: Thanks. We indeed did a survey at the beginning of the course. However, due to technical problems, it was only available after the first day, when we already covered a lot of topics regarding HPC and slurm. We will ask the students to recap the course and give feedback. This may also give some more insights about the longer term benefit of the course. Students can again consider which skills they developed during the course that they are using since. Moreover, the course will be given this year again and a survey will be done before the course starts and at the end of the course.

*AC: We did a survey (retrospectively) with the course participants to assess their skills and knowledge, interests and expectations as well as feedback after the course. We present the results in the new section 3 "Participants' feedback"*

*RC1: Specific comments:*

*RC1: 1. Line 28: By "realistic models," do you mean operational models? The phase "realistic" implies that the results from CM1 are unrealistic, which is not correct.*

AR: Thanks. Yes we meant operational models. We will change the term.

*AC: Done*

*RC1: 2. Line 35: Since the authors are specific in their choice of model, what software was used to visualize model results? This is mentioned later, but would be helpful to describe here as well.*

AR: We will add the choice of visualization models here.

*AC: Done*

*RC1: 3. Line 42: Is visualization instruction rare at your institution, or more broadly? If so, please provide a citation or other reference.*

AR: Generally visualization is taught during the programming introductions. However, 3D visualization, as intended in this course, is not covered in much detail

*.AC: Added the specification 3D visualization to this sentence*.

*RC1: 4. Line 134: I don't think there is enough information given about what is active learning or what sorts of learning activities occurred in the course on a day-to-day basis for the authors to be able to describe the course as using an active learning approach. It may be an appropriate label, but there simply isn't enough information provided to the reader.*

AR: Thanks, we will elaborate.

*AC: We explained the course design and active environment in more detail in section 2.*

*RC1: 5. Line 156: I'm unfamiliar with the term "fazit"*

AR: We will change it to feedback.

*AC: Done*

*RC1: Technical corrections:*

*RC1: 1. Line 16: The phrase "…often essential for a further career in meteorology" sounds a bit off. Further, not every single scientist who earns a degree in meteorology needs these skills. Thus, I suggest rephrasing to something like "…often essential for many career paths in meteorology"*

> AC: Thanks, we will change the sentence according to your suggestion.
>
> *AC: Done*

*RC1: 2. Line 84: "A follow-up course is in planning" should be "A follow-up course is currently being planned."*

> AC: Thanks!
>
> *AC: Done*

**Changes done according to Reviewer #2 comments**

AR We thank Reviewer #2 for the helpful comments! Please note that we will change parts of the paper based on the suggestions of Reviewer #1: We would like to reorganize parts of the paper. We will add more details regarding the course background, design, requirements and goals in the introduction. Additionally, we used a survey to ask the participants of the first course to recap what they have learned. These results will be added to the conclusion part. To stay within the recommended page range, we will shorten the part about the students' contributions and write it in a more concise way. As a result of these changes, some of your comments might not be fully addressed in the final paper. However, we greatly appreciate your valuable feedback, which has contributed to improving the quality of our paper.

Please note that we formatted the reviewer comments in bold and cursive text settings, and the authors' answers in blue.

> *AC: We rewrote parts of the course, especially the introduction where we added learning objectives. We added results from a survey that assessed the acquired skills of the participants. To stay within the page count, we shortened the part on the student's projects, combined figs. 2 and 3 in one figure and moved the written feedback of two participants to the supplementary material.*

**Comments of Reviewer #2 (abbreviated as R2): Recommendation: Minor revisions**

**RC2: Summary: The paper outlines a 2-week block course designed to teach participants how to run the CM1 model and use state-of-the-art visualization tools to analyze the data. Detailed lesson plans are also provided, enabling instructors to easily adopt the material. The paper is well-written, and I only have a handful of minor suggestions.**

> Author Responses (abbreviated as AR) to the Reviewer Comments: Thanks for your helpful comments! We will address you comments point by point in the following.

**General comments:**

**RC2: 1. Can the authors provide a URL pointing directly to the course material? I landed on a German webpage that I found hard to navigate (I struggled a bit finding the course content).**

AR: Thanks for the advice! We additionally uploaded the material to researchgate (under the same license), so it should be easier accessible for non-german speakers. The reference is: Schielicke, Lisa, 2024: "Cloud Model 1 & Visualization - A Block course", ResearchGate, Preprint, January 2024, last access: 2-Feb.2024. DOI: 10.13140/RG.2.2.30017.12642

*AC: We added the reference on Researchgate.*

**RC2: 2. Perhaps use "participant" rather than "student"?**

AR: Thanks, we will change it according to your suggestion.

*AC: Done*

**RC2: Specific comments:**

**RC2: 1. Line 2: CM1 is actually convection resolving (not only convection permitting).**

AR: Thanks, we will use convection-resolving instead

*AC: Done*

**RC2: 2. Line 10: Suggest adding "e.g." before the reference.**

AR: Thanks, we will do this.

*AC: Done*

**RC2: 3. Line 21: Use state-of-the-art instead of up-to-date? And again, consider "cloud resolving" instead of convection permitting.**

AR: We will follow your advice.

*AC: Done (used convection-resolving though)*

**RC2: 4. Line 25: run  documented?**

AR: Thanks, we will change "run" to "documented" in line 24.

*AC: Done*

**RC2: 5. Lines 48, 49: Replace "chapter" with "section"**

AR: Thanks, we will replace chapter with section.

*AC: Done*

**RC2: 6. Line 55: Did the participants have to produce the 20 page report within the 2-week period?**

AR: No, they had approximately 4 weeks to finish the report. We will clarify this in the text.

*AC: Done*

**RC2: 7. Line 67: Here and elsewhere: Instead of single-computer, consider "single-processor" or "serial application"**

AR: Thanks! We will change it according to your suggestion to single-processor.

*AC: Done*

**RC2: 8. Line 80: Is this level of detail needed (background of individual participants)?**
AR: Thanks, we will delete the extra information.
*AC: Done*

**RC2: 9. Line 81: Suggest rewording: Not or only partially covered.**
AR: Thanks, we will follow your advice.
*AC: Done*

**RC2: 10. Line 94: How did the participant quantify the degree of lift in the field?**
AR: Thank you for the comment - indeed our text can be more clear here: With "weaker" forcing, it is meant that storm cell intensity is regarded in a thermodynamic profile associated with a weak forcing scenario. This scenario was characterized by the absence of quasi-geostrophic forcing as well as the absence of frontal lift. As a measure to investigate cell intensity, the participant used the maximum vertical velocity of each cell. In a later work based on the presented work during this course, the participant compared his results to a strongly-forced situation using a sounding associated with frontal lift on another day:

REFERENCE: Sperschneider, A. and Bott, A.: Influence of the Orography of West-Central European Low Mountain Ranges on the Intensity of Deep Moist Convection, 11th European Conference on Severe Storms, Bucharest, Romania, 8–12 May 2023, ECSS2023-89, https://doi.org/10.5194/ecss2023-89, 2023.
*AC: This part is dropped in the main manuscript, but in the supplementary material are more details.*

**RC2: 11. Line 104: Higher-resolution: Compared to what—perhaps report the grid spacing used in the different simulations?**
AR: The majority of the simulations had horizontal grid spacings of 1000 m and a vertical grid spacing of 500 m. In the high-resolution run the horizontal grid spacings changed to 500 m. We will clarify this in the text and add the grid spacing to the figure.
*AC: This part is dropped due to the shortening of the text*

**RC2: 12. Line 111, section header: Instead of "classic" perhaps use "intermediate"? The term "classic supercell" is usually part of the "high-precipitation", "classic", "low-precipitation" trio.**
AR: Probably the text is not clear enough. Indeed, we focus on a classic supercell structure (Weisman-Klemp sounding) and investigate if it evolves into a high-precip or low-precip storm in response to modifications of the environmental moisture profile. We will clarify this point in the manuscript accordingly. However, it might change due to the changes we plan to make in this chapter.
*AC: Due to our comment, we let the title as it was*

**RC2: 13. Line 125: Add a reference here?**
AR: Thanks, we will add a reference here.
*AC: We added a reference here.*

**RC2: 14. Line 157: Replace "Fazit" with "feedback", and consider removing "and co-author to this paper**

AR: Thanks, we will change both according to your suggestions.

*AC: Done*

---

## Author Response (AR2)

**Answers to comments received 20-Mar-2024**

We would like to thank the reviewers and Heini Wernli very much for their helpful comments. In the following we will answer the last comments we received. Our answers are given in blue, the reviewer's and editor's comments are bold, black and cursive. Texts starting with AR (AC) are the author's responses (author's changes). Note that we exchanged Figure 1 with a Latex-Table.

**Answers to Heini Wernli's comments (abbreviated by HWC):**

**HWC: L3: why "clusters" (plural), it seems that you worked on one HPC cluster?**
AR: Thanks
AC: We changed it to cluster (singular)

**HWC: I would find it interesting to have a little bit more information about "Bonna": is this a CPU or GPU machine? Is it really HPC or rather a large Linux cluster? Maybe you can add somewhere a few specifics, or add a link to a webpage that describes the cluster.**
AR: Bonna is a Massively Parallel Processor (MPP) cluster that has been available to scientists at the University of Bonn since 2019. Bonna has 70 MPP Nodes, each featuring dual Intel Xeon Gold 6130 processors running at 2.10GHz, with 16 cores and 16 threads per CPU, with a total of 2,240 cores, total RAM of 13.1 TB and a scratch storage of 428 TB. The operating system is CentOS Linux and the batch system is Slurm. More specifics can be found here: https://www.hpc.uni-bonn.de/en/systems/bonna (last access: 21-Mar-2024).
AC: We added the following sentence to section 1: "Bonna is a Massively Parallel Processor (MPP) cluster with 70 nodes, each with dual Intel Xeon Gold 6130 CPU processors which operates on CentOS Linux with a Slurm batch system."

**HWC: L181: why "??"**
AR: Thanks.
AC: We added the correct label to the section in the supplementary material.

**HWC: L182: not sure that I understand: does the understanding help the visualisation or the other way round?**
AR: It was meant the other way around: "understanding through visualization".
AC: We changed the sentence to: "Data visualization helped to deepen the understanding of convective processes."

**HWC: L187: This sentence is repetition from a few lines above**
AC: We deleted the sentence in line 187 and added a reference to the first sentence of the conclusion.

**HWC: you have three different spellings of "Slurm", please decide for one!**
AR: Thanks.
AC: We decided to use "Slurm" throughout the paper.

**Answers to the Comments of Reviewer #1 (abbreviated with RC1)**

**RC1: Line 70: Change "10 days period" to "10-day period"**

Done.

**RC1: Line 74: "how to get started and consecutively evolve the abilities" is grammatically awkward; recommend something like "how to get started and continuously improve their modeling skills"**

AR: Thanks.

AC: We revised the sentence as follows: Participants learned not only the basics of how to work with a numerical model but also how to get started and continuously improve their modeling skills.

**RC1: Line 81: "course contents were" should "course content was"**

AC: We changed it to "course content was".

**RC1: Line 83 and elsewhere: "google" should be "Google" since it is a company**

AC: Done. Thanks!

**RC1: Line 105: Remove the comma after "Note"**

AC: Done.

**RC1: Lines 125 to 127: Change "In order to see whether the corresponding goals were meaningful 125 on the one hand and also acquired during the course period on the other hand" to something like "To see whether these learning goals were both meaningful to and achieved by the participants..."**

AC: Thanks, we revised the sentence according to your suggestion.

**RC1: Line 142 and elsewhere (e.g., Line 180): Modify "significantly" to "substantially"...use of "significant" implies the result was statistically significant, which isn't possible with the data you have**

AC: Thanks, we changed it to substantially and substantial in line 180.

**RC1: Line 181: Remove double question marks**

AC: Thanks, we removed the double question marks and added the correct reference to the section in the supplementary material.